# Multiple-Resolution Tokenization for Time Series Forecasting with an Application to Pricing

## Abstract

We propose a transformer architecture for time series forecasting with a focus on time series tokenisation and apply it to a real-world prediction problem from the pricing domain. Our architecture aims to learn effective representations at many scales across all available data simultaneously. The model contains a number of novel modules: a differentiated form of time series patching which employs multiple resolutions, a multiple-resolution module for time-varying known variables, a mixer-based module for capturing cross-series information, and a novel output head with favourable scaling to account for the increased number of tokens. We present an application of this model to a real world prediction problem faced by the markdown team at a very large retailer. On the experiments conducted our model outperforms in-house models and the selected existing deep learning architectures.

## 1 Introduction

Forecasting remains a frontier for deep learning models. The value of improved forecasting capabilities is obvious as it enables us to improve the contextualisation of decision making. The model we propose was specifically developed for time series for which we have control over some of the auxiliary variables. We assume those auxiliary variables have a significant effect on the time series. Since auxiliary variables are hard to model with traditional statistical approaches we turned to deep learning. We specifically focus on pricing problems: a canonical example in which the auxiliary variable of price affects sales. The ubiquity of pricing problems and resulting large datasets make them suitable for deep learning models. The forecasting problem in pricing is suitable as it benefits from being able to aggregate many similar individual pricing time-series across stock keeping units (SKUs) and stores. The model we propose is based on the celebrated transformer architecture in which self-attention is leveraged to explicitly model pairwise relationships between tokens. We believe this choice of architecture is justified by the complexity of the underlying process of consumer behaviour in response to changes in price and large dataset sizes.

This work is about tokenisation strategies for time series and aligning a transformer architecture with these strategies. We believe that a key way of advancing transformer capabilities for time series forecasting is by developing and validating specialised tokensiation strategies. Our contributions are:

- A set of new tokenisation modules to obtain: multiple-resolution (MR) past data tokens, MR known time-varying tokens, and cross-series tokens. We differ from existing multiple-resolution approaches as all tokens are passed to a single attention mechanism creating a larger context window.

- A reverse splitting output head which encourages output token specialisation and with favourable scaling properties to deal with the increase in the number of tokens.

- An application of the model to a noisy real-world forecasting problem faced by the pricing team at a very large retailer and ablations on an analogous problem based on a public retail dataset. Our method outperforms existing in-house methods and two popular existing architectures on a both problems.

## 2 Deep Learning for Time Series Forecasting

Denote the forecast horizon as $f$ and the lookback horizon as $l$. We define three forms of data associated with each individual time series:

- Static $\mathbf{s}$ (Store information, product information),

- Time-varying known $\mathbf{x}_{t_0-l+1:t_0+f}$ (global seasonal data, prices),

- Observed data/time-varying unknown $\mathbf{y}_{t_0-l+1:t_0}$ (sales, weather)

where $t_0$ is the point at which we want to make our forecast. The multivariate forecasting problem at $t_0$ with a forecast horizon of $f$ is the prediction of future values of the series for the time periods $t_0 + 1$ to $t_0 + f$ across all channels $c_1, ..., c_n \in C$ which we denote as $\hat{y}_{c_1:c_n,t_0+1:t_0+f}$ to minimise some loss function $loss(y_{c_1:c_n,t_0+1:t_0+f}, \hat{y}_{c_1:c_n,t_0+1:t_0+f})$. Each channel respectively corresponds to each variate we are interested in forecasting simultaneously. Channels are a choice reflecting our structural view of the time series of what should be predicted simultaneously for a given context. Larger training sets are produced by concatenating series and adding a static variable denoting some characteristic of the series.

### 2.1 Transformers

Components of the original transformer architecture (Vaswani et al., 2017) powers the state of the art for large language models and has seen more computational resource investment than any other deep learning architecture. Empirically, the transformer is outstanding for language modelling and benefits from scaling laws. Due to the success of transformers with the sequential task of language there has been considerable interest in applying transformers to time series. Academic research has produced mixed results and tends to be limited in the scope of study.

Transformers have architectural limitations when applied to time series (Zeng et al., 2023). Transformers are permutation invariant and thus inherently do not create an inductive bias which reflects that observations happen sequentially. Transformers for language use two core architectural adjustments for dealing with the lack of sequential inductive bias. Positional encoding adds a bias to the latent matrix representation of a language sequence which is designed or learnt to reflect sequentiality. Decoder architectures mask out the attention matrix so that each unit of latent representation (token) can only observe the tokens that occurred before it in the sequence. The effectiveness of either of these techniques for forecasting is still an open question. The evidence so far is mixed and simple models have been shown to outperform very complex transformer models on academic benchmarks (Zeng et al., 2023).

At a higher level, the concept of a token in language[1] does not trivially map to time series. Time series are composed of a sequence of observations over time but single observations are not necessarily meaningful units in terms of the dynamics of a time series. An analogous reasoning from classical time-series approaches is the concept of the order of integration which is the number of times a time series has to be differenced (how many adjacent points need to be considered) to become stationary. Time series are often decomposed into trend and seasonality or frequencies, both of which are transformations of more than just a single observation. A notable transformer time series architecture uses frequency decomposition to tokenize a time series (Zhou et al., 2022). Alternatively, a popular and increasingly standard method (Das et al., 2023) for tokenisation is patching (Nie et al., 2023): it splits a series up into fixed length sequences which are then mapped into a latent space using a shared learnable linear transformation. The original PatchTST uses patches of length 16 with a stride of 8 allowing for overlapping patches. This can be interpreted as the resolution (number of observations informing a token) at which a time series is processed. As opposed to single period to token designs (Lim et al., 2021) patching loses the 1-to-1 correspondence between tokens and forecast horizon periods. Instead, the matrix output from the transformer blocks is flattened and passed to a large linear layer (presenting a potential bottleneck) which outputs the forecast vector. The implicit assumption is that

---

[1]Note that tokenisation in LLMs is typically not trained end-to-end and is fixed, transferred into the model from a different pretraining task. This is often a cause of problems.

meaningful information about the time series occurs at higher resolutions than the individual observations in the time series. A linear model over the whole series assumes that we need to look at the highest resolution: the whole series. Our work builds on this assumption by extending patching to multiple resolutions.

There have been many attempts to design transformer and transformer like architectures for time series, particularly for long forecast horizons due to a set of popular testing benchmarks. The vast majority of innovation appears either in the design of the tokenisation or the architecture of the token processing engine. This repository is an excellent overview of recent developments across a range of modelling approaches.

## 3   The Model: Multiple-Resolution Tokenization (MRT)

Our model is based on two ideas: that time series carry relevant information at multiple resolutions across all associated data and that all of these relationships should be modelled explicitly. The latter enables us to extract conditional leading indicators which only appear for some subset of series in a large corpus. Transformers are a natural choice for such modelling as they enable us to explicitly model pairwise relationships between tokens. Defining meaningful units of information has been a key question in the development of time series. In this context tokens should be understood as useful representations of some facet of a time series. In designing separate tokenisation for past and auxiliary data we implicitly assume that auxiliary data carries separately significant information. Designs which include auxiliary information often meld all information at a given token together (Lim et al., 2021; Chen et al., 2023), whereas we process separately for each type of auxiliary information. The separation into more tokens enables explicit attention links to aspects of auxiliary information which increases interpretability. The multitude of resolutions complemented by representations of auxiliary variables as a tokenization strategy enables the learning of complex pairwise leading representations.

To add clarity we introduce a common notation to describe the view of the tensor and the operation performed on it at a given point in the model. We will describe the dimensions of a tensor with a collection [   ] of letters which represent dimensions. The last $k$ letters represent the dimensions the operation is taking place over. Most operations are performed over a single dimension, however there are exceptions for example self attention operates on matrices so the last two dimensions. If a tensor has been flattened this will be indicated in the new view as with a · and if two tensors have been concatenated in a dimension we will use +. The relevant dimensions in our case are $B$ for batch or sample, $C$ for channel, $T$ for time and for patch which is typically a compression across the time dimension and auxiliary variable embeddings (can be understood as the token dimension), $V$ for the variable dimension, and $L$ for the latent dimension. Note that these dimensions will not describe unique states as they serve as illustrators as opposed to an exact functional description of the model which can be found in the code. We use lower case letters for fixed values to emphasise the size of the dimension, for example (unless stated otherwise) the size of the latent dimension is $d_m$ so $[B, C, T, L = d_m]$. To illustrate this notation consider the standard output head of PatchTST (Nie et al., 2023), the operation takes the output from the transformer in the form $[B, C, T, L]$, flattens it to $[B, C, T \cdot L]$ applies a linear layer to the last dimension to produce an output $[B, C, f]$. In short we denote this as: linear layer $[B, C, T \cdot L] \rightarrow [B, C, f]$. In the case of linear layers we can also identify the size of the matrix that needs to be learnt in the transformation. For a generic linear transformation $[..., X] \rightarrow [..., Y]$ the learnt transformation matrix $\in \mathbb{R}^{|X| \times |Y|}$.

We employ channel independence in our base and auxiliary architecture, which is a broadly used modelling approach in the field. Channel independence means that the computation never occurs across dimension $C$ and the channels are treated as independent samples by the modules. We propose a separate module which extracts cross-series information in the form of cross-series tokens.

### 3.1   Multiple-Resolution Patching

The multiple resolution patching is defined by a resolution set $K = k_1, ..., k_r$ takes an input of the form $[B, C, T = l]$ and uses a set of $|K| = r$ linear transformations to create an output of the form $[B, C, T, L]$. Denote the size of the latent dimension of the model as $d_m$. We extend patching to multiple resolutions by using several divider blocks. A divider block splits a time series into $k_i$ roughly equal parts. For a series

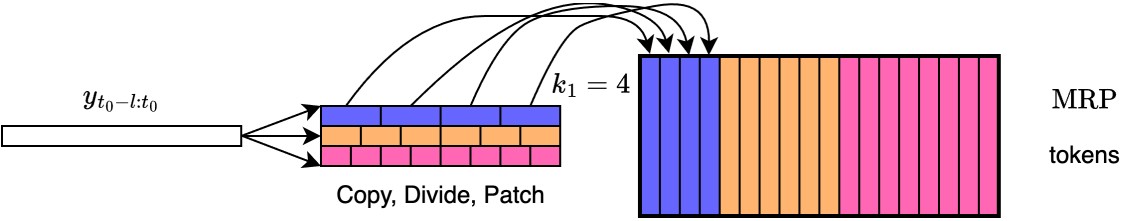

Figure 1: Multiple Resolution Patching (MRP)

of length $h$ the base patch length is set as $b_i = \lfloor h/k_i \rfloor$. If $k$ does not divide $h$ we increase the length of the first $h - b_i k$ patches by 1. In our application $h$ is either $l$ (past data) or $l + h$ (time-varying known). For each resolution $k_i$ we learn a linear projection into the latent space $\mathbb{R}^{b+1} \mapsto \mathbb{R}^{d_m}$. Weights are shared for all patches within a resolution. We left pad patches that are $b$ long with a 0. The core operation is $[B, C, T = b_i + 1] \to [B, C, L = d_m]$ which is repeated $\sum_{k_i \in K} k_i$ times ($k_i$ times for each resolution) and stacked along a patch/token dimension into $[B, C, T = n_{MRP}, L]$. The created dimension $T$ contains $\sum_{k_i \in K} k_i = n_{MRP}$ tokens obtained from multiple resolution patching. An illustration of multiple resolution patching is presented in figure 1.

### 3.2 Handling Auxiliary Information

A key advantage of deep learning models over classical models is that they can accommodate arbitrary forms of input enabling us to jointly model all available information for a time series. Auxiliary information is not trivially useful for time series forecasting. Most recent state of the art papers exclusively focus on past/observed data. The inclusion of auxiliary data often leads to issues with overfitting, explainable by the noise in time series data and small datasets. In existing work which includes auxiliary information it is typically mixed with past data to form tokens representing all information at each time step or resolution. We take a different approach and design a separate tokenisation scheme for different types of auxiliary data. This enables us to explicitly learn contextual representations of the time series that are purely auxiliary and not directly dependent on the noise in the time series, which should help reduce overfitting.

**Base Tokenization**  Data for time series is either continuous or categorical. Categorical variables need additional processing, in the case of transformers entity embeddings are standard practice and amount to learning a vector embedding for each type within a category. To match the embedding size in the case of numerical auxiliary variables we employ numerical embeddings. We learn a vector for each numerical auxiliary variable which is then scaled by the value of that variable. We call this process base tokenisation. To handle missing values, we assign a separate learnable vector embedding for each variable in cases the value has not been observed. When there is variation in the length of the input horizon we left pad all temporal series with a special category or numerical value that always yields a zero vector embedding, which then does not activate the corresponding aspects of the network. We do not utilise these base auxiliary tokens directly as the inclusion of many auxiliary variables, especially time-varying known tokens (TVKT), would dominate the transformer input and the increased input size would scale poorly due to the quadratic complexity of self-attention. In multivariate forecasting, the auxiliary data needs to be distinguished as global (ex: holiday dummies) or specific (ex: price data). In the case of global variables the values for a given sample are identical across $C$. This distinction depends on the choice of aggregation. In our implementation use the same module architectures to process both specific and global auxiliary variables but implement them separately generating one group of specific and global TVKT.

We include auxiliary variable tokens $[B, C, T = n_S + n_{TVK}, L]$ where $n_S$ is the number of static tokens and $n_{TVK}$ is the number of TVKTs by right concatenating them with the output of the MRP to obtain $[B, C, T = n_{MRP} + n_S + n_{TVK}, L]$.

**Time-Varying Known Variables (TVK)**  The TVK of a time series form a tensor $[B, C, T = l + f, V]$. We process global and specific variables separately. The base tokenisation first transforms both the

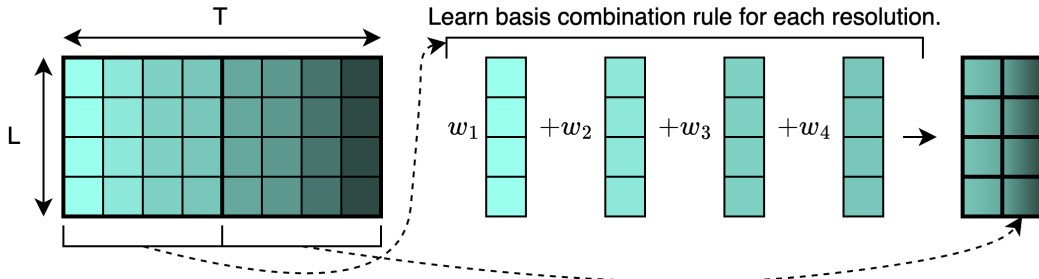

Figure 2: Basis combination operation as used for tokenising time-varying known variables.

categorical and numerical variables in the tensor from $[B, C, T, V] \rightarrow [B, C, T, V, L]$. We then learn a linear projection that compresses the variable dimension to one $[B, C, T, L, V] \rightarrow [B, C, T, L, V = 1] = [B, C, T, L]$. This base tokenisation plus mixing across variables creates one auxiliary token per time step.

In line with previous multiple resolution patching we apply a modified version built on basis combination. In contrast with past-data we want to patch across tokens (collection of vectors) not individual observations (collection of points). We split the tensor $k_i$ roughly equal parts for each resolution $i$ along the time dimension. We use the same resolutions, rules on length (denote the length of a patch $a_i$), and padding (so each patch is actually $a_i + 1$), but for a longer series as the size of $T$ is $l + f$. The splitting leaves us with a number of tensors of the shape $[B, C, L, T = a_i + 1]$. We treat the columns of each matrix in the last two dimensions of the from $[L = d_m, T = a_i + 1]$ as a basis. For each resolution we learn a basis layer: vector of size $a_i + 1$ which linearly combines the columns of the matrix into a single vector. The basis layer transforms $[B, C, L, T = a_i + 1] \rightarrow [B, C, L, T = 1] = [B, C, L]$ for each patch in each resolution. This operation is illustrated in figure 2. The basis combination results in $\sum_{k_i \in K} k_i = n_{MRP}$ such tokens. Note that $n_{MRP}$ at this stage is equal for both past data and TVK data despite the two sequences having different lengths as our resolution set is defined in terms of the number of parts a sequence should be split into which corresponds to the number of tokens produced by each resolution. The tensors are stacked along a patch dimension into $[B, C, T = n_{MRP}, L]$. To reduce the relative influence of auxiliary variables and reduce the size of the input matrix to the transformer we employ a linear compression layer to compress the patch dimension to a predefined number of tokens $n_{TVK}$. This layer mixes across patch representation. The transformation is a linear layer $[B, C, L, T = n_{MRP}] \rightarrow [B, C, L, T = n_{TVK}]$, which we transpose to $[B, C, T, L]$. This process creates a a set of TVKT which are a mixture of representations extracted at multiple resolutions from joint base representations of the time-varying auxiliary data.

**Static Variables** Static variables form a tensor $[B, C, V]$. They do not have a time component and are equal across channel $C$ if they are global and can differ if specific. The base tokenisation transforms the static variable tensor into $[B, C, V, L]$. If $|V|$ is low we append the base tokenisation to the MRP directly meaning the number of static tokens (ST) $n_S = |V|$ and the variable dimension is just treated as the token dimension. Otherwise we employ an additional linear layer $[B, C, L, V] \rightarrow [B, C, L, T = n_S]$ which operates over the $V$ dimension and condenses the representation to a predefined number of mixed tokens $n_S$.

### 3.3 Cross-Series Information

The overfitting problem has also been empirically observed for the inclusion of cross-series information, which has motivated the use of channel independence Han et al. (2024). We propose a channel mixer module which employs a mixer architecture to generate a set of cross-series tokens (CST) from the tensor containing all other tokens.

We call the module responsible for extracting cross-series information the channel mixer. As input the channel mixer takes the tensor of all existing tokens MRP, ST, and TVKT. Define the number of these tokens as $n_B = n_{MRP} + n_S + n_{TVK}$. The input tensor takes the form $[B, C, T = n_B, L]$. We employ a mixer architecture inspired by Chen et al. (2023) to extract cross series tokens. Mixer architectures have been

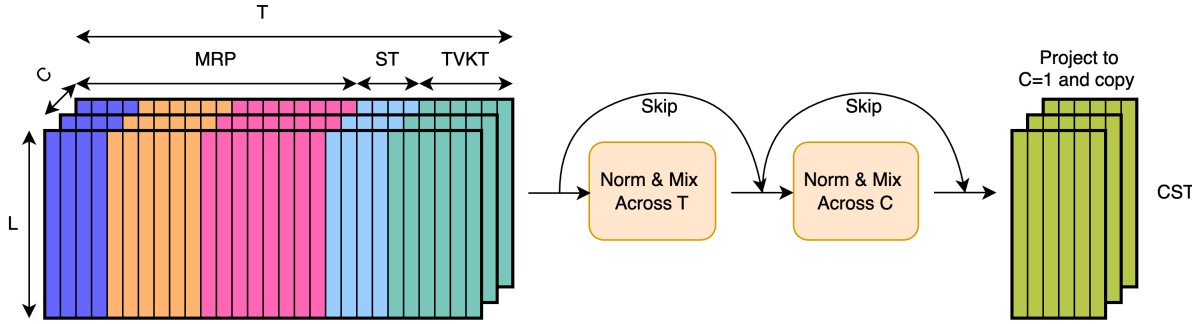

Figure 3: Channel mixer: a module for learning cross-series tokens.

shown to perform well on time series forecasting while extracting cross-series representations in every block. We roughly illustrate the module in figure 3. The operating tensor is normalised before each mixing step and a skip connection is employed across each mixing step. The first mixing step is across the token dimension $T$, applying a linear transformation $[B, C, L, T = n_B] \to [B, C, L, T = n_B]$ followed by an activation function and dropout. The number of tokens is then squeezed down to a predefined number $n_{CST}$ with another linear layer $[B, C, L, T = n_B] \to [B, C, L, T = n_{CST}]$ (we use $A$ here to signal that the tokens have been mixed into an auxiliary information representation). The second mixing step is across the channel dimension $C$. Two linear layers are employed, the first projects the channel dimension of size $d_c$ into the latent dimension of the cross series module $d_{cross}$ (adjustable hyperparameter), so $[B, A, L, C = d_c] \to [B, A, L, C = d_{cross}]$. After an activation function and a dropout layer the second linear layer projects back into the original channel dimension $[B, A, L, C = d_{cross}] \to [B, A, L, C = d_c]$. After the end of the second mixing another linear layer squeezes the channel dimension to one leaving us with $[B, C = 1, T, L]$. The reason this is done is so that the cross-series tokens are the same across channels. To align it with the dimensions of our collection of tokens we repeat the tensor $|C| = d_c$ times to obtain $[B, C = d_c, T = n_{CST}, L]$. We append it to all other existing tokens to obtain the final input matrix $A$ with size $n_{MRT} = n_B + n_{CST}$ which is passed to the transformer module $[B, C, T = n_{MRT}, L]$.

### 3.4 Output Head

We propose reverse splitting, a novel output head which has favourable scaling properties. The transformer blocks produce a matrix which is the same size as the input matrix $[B, C, T, L]$. Using the same resolution values as for the splitting, we reverse the process. For resolution $k_i$ we take the next $k_i$ column vectors from the output matrix corresponding to the input positions of the MRP tokens and individually project them into patches that are then composed to represent the forecast. The length of the projected patches ($p_i$ or $p_i + 1$) is determined the same way as the splitting in MRP. We learn a projection matrix for each resolution $\mathbb{R}^{d_m \times p_i + 1}$ and omit the final vector value for the reverse patching for which we have set the length to $p$. The core operation iterates over the dimension $T$ so each input is $[B, C, L]$ which is projected into $[B, C, T = p_i + 1]$, this is repeated $k_i$ times and concatenated according to the length rule into $[B, C, T = f]$. This leaves us with $|K|$ forecast vectors of the form $[B, C, T = f]$ which we simply sum to obtain the final output. We call this reverse splitting. We illustrate this module in figure 4. It scales favourably to simple flattening as it requires us to learn $d_m p_i$ parameters for each resolution $i$ for a total of $d_m \sum (p_i + 1)$. This means that scaling depends on a sum of fractions of the forecast horizon $f$ and inversely depends on the resolution (number of divisions) as $p_i \approx f/k_i$. Restated, our output head requires approximately $d_m f \sum \frac{1}{k_i}$ parameters. In comparison, the flattening approach would require learning $d_m f \sum k_i$ parameters which is considerably greater as $k_i$ are natural numbers. Note that the reverse splitting does not use the corresponding position outputs of auxiliary tokens and can be applied even if auxiliary information is not included. In addition to favourable scaling, the design of reverse splitting was motivated by enabling the learning of forecast mappings at different resolutions. These resolutions have a symmetry with MRP patching and enable the model to learn a multiplicity of output mappings increasing predictive robustness similarly to ensembles. Moreover, output tokens are specialised to predict a segment of a specific length at a specific position so each output

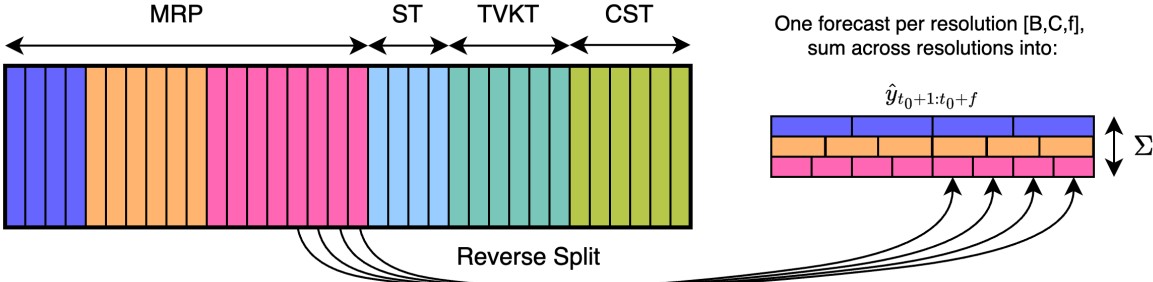

Figure 4: Reverse splitter - the output head only operates on post transformer tokens corresponding to the original MRP positions.

token explicitly serves a different purpose. This is in contrast to typical output head designs in which the output tokens are flattened and passed to a single linear layer meaning there is representational multiplicity.

### 3.5 Other Architectural Choices

The tensor of all tokens $A$ is processed by a sequence of transformer blocks after which it is passed to the reverse splitter output head. We use multi-head self-attention in the transformer block. The learning in self attention can be interpreted as learning four different mixing rules across the $L$ dimension to infer good pairwise comparisons. Since the focus of this work is on tokenisation strategies we avoid extensively investigating positional encoding or masking strategies.

**Masking**  We use an encoder design without masking. While a decoder architecture with masking could be designed to be time consistent with patching at multiple resolutios it is not trivial to extend it to TVKT and CST so we avoid masking. Any form of mixing such as compressing TVKT or CST across the $T$ dimension obscures causality.

**Positional Encoding**  We use learnable positional encoding. This is equivalent to learning a fixed bias for the input matrix to the transformer. We do this as it is difficult to design positional encoding for a complex set of tokenisation procedures. Certain patching approaches use aggregations over conventional positional encoding, but that is still limited to only past data.

**After Attention**  We follow the standard transformer architecture: self-attention is followed by a skip connection, normalisation and then a feed-forward network. We use a standard two linear layer MLP with an activation function and dropout. The latent dimension in between the two layers is another hyperparameter $d_{ff}$. The feed-forward network is applied to the latent dimension, so separately to each token. The operation is $[B, C, T = n_{MRT}, L] \rightarrow [B, C, T = d_{ff}, L] \rightarrow [B, C, T = n_{MRT}, L]$. This is followed by a skip connection from the input to the feed-forward network and another normalisation.

### 3.6 Limitations

Multiple resolution patching and associated auxiliary variable tokenisation adds many more hyperparameters in the form of the resolution set. The space of resolution sets is combinatorial making hyperparameter optimisation a challenging task. We somewhat constrain the space by not allowing overlaps, thus avoiding the need for a stride hyperparameter associated with every resolution. Our tokenisation modules increase the number of learnable parameters in the embedding especially for small $k_i$ which makes scaling to large latent sizes more difficult. The CST are sequentially computed downstream of all other tokens limiting the degree of parallelisation in tokenisation. All tokenisation is done end-to-end a degree of overfitting is expected especially with CST tokens as they are computed using all other tokens.

Computationally the main difficulty is the increased number of tokens which scales computation quadratically. This constrains how high $n_{MRP}$ can get limiting dense resolution sets (high $k_i$) which we hypothesise are useful as longer context windows for problems with long lookback horizons. Our novel output head design provides a counterbalance in terms of scaling as it is much more computationally efficient than flattening output heads which tend to consume the bulk of computation in models like PatchTST. Exploiting and encouraging sparsity in self-attention may be one possibility to deal with the larger context window. Current production large language models are designed to handle context windows which are three to five orders of magnitude higher than what we demonstrate in our experiments.

## 4 Existing Multiple Resolution Approaches

We would like to outline the differentiating contributions of our work in contrast with other recent approaches which employ the idea of utilising multiple resolution modelling in transformer architectures. We provide a brief description of each model and point out the key difference in capabilities and architecture. Our model is unique in that it uses the tokens from multiple resolutions in the same attention mechanism allowing for explicit pairwise modelling at multiple resolutions across multiple data types and uses a completely novel output which encourages every relevant output token to assume a specialised role. In addition we provide a scheme for how to tokenize time-varying known data at multiple resolutions including for the time steps which occur in the future.

**Shabani et al. (2022)** The Scaleformer was the first notable attempt at multi-scale modelling applied to time-series forecasting with transformers. It iteratively implements broader pooling which forms the inputs for an encoder and feeds the decoder linear interpolations of the prediction of the previous layer. The outputs of all layers are combined with a "normalising"-weighted sum. It does not process multiple resolutions simultaneously, instead opting to do so sequentially. It only handles past/observed data and not any auxiliary variables.

**Chen et al. (2024)** The Pathformer is designed to work as a foundation model, amenable to working with time series of differing scales. It is composed of many blocks which operate at different resolutions. Each block first uses QKV-attention to process by individual patch, allowing for cross-series modelling and the inclusion of all data available at a given past time-stamp by compressing it into a single vector patch embedding. The next step is a layer of self-attention which models the pairwise relationships of all of the resulting patch embeddings. This is paired with an adaptive pathway which routes the input series into a weighted combination of the resolution blocks. The routing is based on a learned embedding obtained from the trend and seasonality of a series. At no point does attention look at multiple resolutions simultaneously meaning the routing is the only source of modelling such relationships which is sensible when trying to accommodate vastly different time-series.

**Woo et al. (2024)** Another foundation model. The key motivation is being able to process any arbitrary time series, this approach can model all auxiliary variables. All related inputs are padded, flattened, and concatenated into a single series which is then patched to tokenize. Multiple scales are not used actively. Instead multiple resolutions are defined and contained in the model but only one is used for a given forecasting task. The choice of resolution is determined heuristically.

**Zhang et al. (2024)** This work is the closest to ours in to what degree it models the relationships between embeddings at different resolutions. Each block uses separate branches to patch for different resolutions but at the end of the block it learns a linear projection which fuses all tokens to form a vector which then gets patched again in the next block. Every fusion models the relationships of representations at different resolutions. Interestingly the patching is done at each block. The approach does not extend to auxiliary data or model cross-series relationships.

**Note on Other Multi-Scale Approaches** There is a host of other modelling approaches which use a multiplicity of some input to extract and process representations. Multi-fidelity modelling (Perdikaris et al., 2017) has been used to infer relationships between cheap to obtain low fidelity data and expensive to obtain

high fidelity data. The main interpolation engine used are Gaussian processes which are well suited as universal approximators but have cubic complexity in the number of samples rendering them impractical for large scale problems. They are used in artificially enhancing fidelity in physical simulation. Neural operators (Kovachki et al., 2023) learn maps between continuous infinite-dimensional functional spaces taking the idea of multi-fidelity modelling to the limit as they can take any discretization/level of fidelity as an input and can be queried at any level of discretization for an output. A canonical application is learning surrogate maps for the solution operators of PDEs and weather forecasting Kurth et al. (2023). The capability is impressive but adds an unnecessary level of complexity as output discretization is fixed in our problem. In terms of input fidelity levels, we artificially induce multiple levels of fidelity through tokenisation and have control over this as a hyperparameter so being fully flexible at the input level is not necessary. The fixed structure of the forecasting problem of regularly spaced observations at a single frequency means the additional capabilities of multi-fidelity modelling are not necessary and would add an undesirable level of complexity.

Mixed-Resolution approaches partitions the input into a single set where the partitions can be of different sizes. The partitioning in vision applications (Ronen et al., 2023) is determined by a saliency algorithm. So long as resolution sizes match all partitions present in a mixed-resolution approach will also appear in MRT. Mixed-resolution can be interpreted as a masked MRT with specific constraints on the masking such as the unmasked partitions composing exactly one input according to a saliency score. While this preselection is interesting, it assumes that we should only extract information at a single resolution for each partition of the input limiting the representations we can extract as relevant dynamics may exist between pairs of less salient frequencies.

## 5 Application to a Forecasting Problem in Pricing

### 5.1 Real Markdown Problem

Markdowns are a subset of pricing problems concerned with reducing prices to clear expiring stock. The decision problem is to set a series of price reductions such that an objective is optimised (Aleksandar Kolev & Arafailova, 2023). The objective is typically revenue combined with some sustainability based measure. A key aspect of such problems is predicting the sales of an item in response to a set price in a given context. A forecasting model which incorporates price as an auxliary variable can be effectively be utilised as a simulator to narrow down pricing strategies for empirical testing. There is considerable operational and reputational risk in evaluating previously untrialled strategies. A good simulator enables a virtual exploration of the pricing strategy hypothesis space which can help select potentially more profitable and sustainable pricing strategies for testing.

The data for the experiments is real markdown data from a very large European retailer. The markdown pricing prediction problem is to estimate the sales of an item given a series of past sales, prices and other contextual information. This prediction problem is understood as particularly difficult. The theory is that each time there is a price reduction we observe a spike in sales which then decays. Subsequent price reductions produce increasingly dampened spiking. This highly non-linear behaviour is difficult to model with simple approaches. This is further exacerbated by the high levels of noise associated with consumer behaviour owing to stochastic factors like the weather or competitor behaviour and the human factor in actually delivering the pricing strategy in stores. Historical data is limited to a narrow set of tried and tested pricing strategies such as 25% off then 50% then 75% adding to the epistemic uncertainty. The downstream decision problem of setting reduced prices for items that are about to expire sees about a million decisions made every day.

We set our forecast horizon to $f = 24$ and adjust the lookback horizon accordingly with the longest sequence in a given experiment. We use only two channels: full price and reduced price sales which we attempt to forecast simultaneously. Our model produces 52 tokens. We include more details about the experiment in the appendix.

### 5.1.1 Experiments

The experiments were defined by the range of year-weeks they contained. The experiments are limited to just one week due to memory constraints. Each experiment contains the markdown data for one week across

Table 1: Experiment week 30 with $l = 28$.

| Model | MSE | MAE | full price MSE | full price MAE |
|---|---|---|---|---|
| internal | NaN | NaN | **0.0821** | **0.1443** |
| MRP | **0.0717** | **0.1323** | 0.0895 | 0.1447 |
| PatchTST | 0.0765 | 0.1349 | 0.0963 | 0.1480 |
| DLinear | 0.0742 | 0.1380 | 0.0959 | 0.1549 |

Table 2: Experiment week 45 with $l = 37$.

| Model | MSE | MAE | full price MSE | full price MAE |
|---|---|---|---|---|
| internal | NaN | NaN | 0.0764 | 0.1214 |
| MRP | **0.0269** | **0.0661** | **0.0474** | **0.0877** |
| PatchTST | 0.0660 | 0.1190 | 0.0926 | 0.1370 |
| DLinear | 0.0521 | 0.1156 | 0.0745 | 0.1344 |

all shops and all items. For any experiment the series corresponding to the last 20% (rounded up) of the due-dates were held out as the testing set, the 15% (rounded up) of due-dates immediately preceding were used as the validation set and the remaining as the training set.

We display the results of our two main experiments in tables 1 and 2. The lookback horizon in the week 30 experiment is 28 and 37 in week 45, the forecast horizon is fixed at 24, so the next 12 hours of operation. The week 45 experiment contains approximately 63 thousand series and training took approximately 2 hours per epoch run locally on a CPU. Early stopping in the week 30 experiment was triggered after 7 epochs and after 19 epochs in the week 45 experiment. We only had access to internal predictions on full price sales, so we exhibit the mean squared error (MSE), and mean average error (MAE) for that channel, comparing the internal result with our MRT result. The specifics of the internal predictions methods are confidential but are built with statistical and classical machine learning techniques. We also train PatchTST (Nie et al., 2023) as the canonical transformer for forecasting (patch length 8 with stride 4, same latent dimensions) and DLinear (Zeng et al., 2023) as an example of a simpler alternative to transformers under the same training conditions.

The model is competitive with internal predictions and an improvement on existing models in the week 30 experiment and significantly outperforms all other models on week 45. Note that the test scores have been reverted to their original scales. Given the complexity of the prediction task across a range of products and locations and the necessary adjustments to fit a sequence-to-sequence model the result for week 45 provides strong evidence that MRT is a powerful model which could deliver much with hyperparameter optimisation and scaling. The week ranges on these experiments are somewhat limited but we are confident that over longer ranges these models would be even more competitive due to stronger auxiliary embeddings and general scaling effects.

One contrarian explanation of week 45 results is that many values are 0 and our network learns to predict near 0s in many cases. The many values being 0 case is somewhat credible that the dataset only contains longer series; the longer duration could reflect that getting rid of stock in those cases is difficult. An encouraging sign is that the predictions for reduced price sales are better than those for the full price sales (assuming similar scales across channels). Since we have shown that we are competitive or better than internal models for full price sales, the reduced price sales predictions show even more promise.

### 5.2 Favorita Dataset Problem

In addition to experiments with private data we replicated a markdown type forecasting problem on a public dataset. We use the Favorita dataset (Corporación Favorita, 2017) which contains the daily sales of a range of

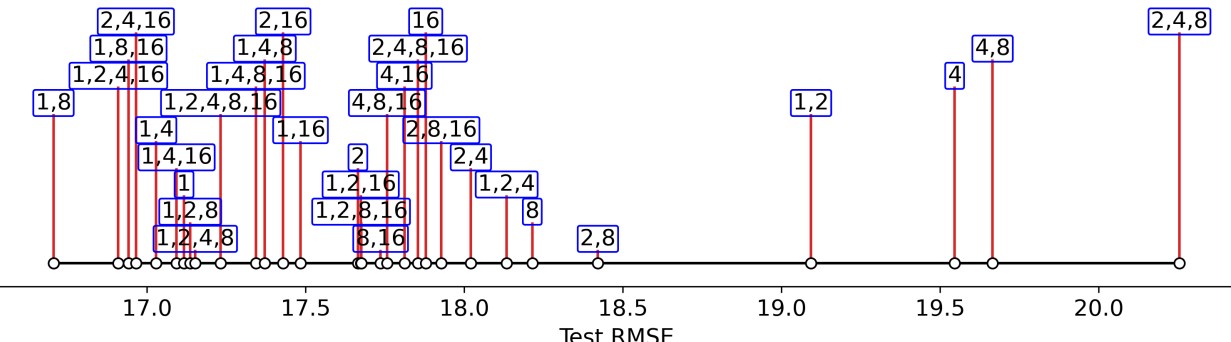

Figure 5: Results on holdout set of resolution set ablation.

items across a range of locations from a South American grocer. In addition it includes auxiliary information such as whether an item was under promotion which is somewhat analogous to the markdown problem. We narrow the dataset to only include the 78 items present in all locations and sold on all dates. We set $f = 16$ and $l = 32$ and apply a sliding window approach to create unique series. We select locations as our channels. This means that for each input we simultaneously predict the sales for the next 16 days for a given item in all 53 shops. The implied assumption from a cross-series perspective is that we believe that there may be leading indicators in some locations for other locations. We use the same holdout approach and base hyperparameters as in the first experiment unless noted otherwise. We conducted three sets of experiments to ablate the model and empirically establish the usefulness of modules.

### 5.2.1 Resolution Set Sizes

Picking a resolution set involves selecting from exponentially many options. We restrict possible resolution sets to including any nonempty combination of $k \in \{1, 2, 4, 8, 16\}$, so 31 possible combinations. We train each model once due to the large number of experiments and acknowledge that there is a significant variance in outcomes. Note that both TVKT and CST were included in these experiments. The results are presented in figure 5. We find that the best performing configurations are ones with more than one resolution, but that more resolutions does not necessarily mean improved performance. The optimal resolution set for a problem depends on the structure of the underlying process. Interestingly the best performing single resolution model is with $k = 1$, which means extracting a single token to represent all past data. The results here strongly support using multiple resolutions, but that the choice of resolution requires careful selection. We pick the second best performing resolution set $\{1, 2, 4, 16\}$ as a baseline in the following ablations.

### 5.2.2 Module Inclusion

We test four variants of the model: no auxiliary variable tokens, just TVKT, just CST, and both groups. In addition we test the two models we compared MRT with in the previous experiment. We run each experiment five times and display the results with standard deviations in table 3. The best performing performance on average was achieved by our MRT model without additional tokens. This further supports the use of MRT, though the gain in performance is slim in exchange for about an order of magnitude increase in computation when comparing it to PatchTST. As expected, including auxiliary variable tokens induces a lot of variance in results due to potential overfitting, but expect that this could be reduced by balancing the number of auxiliary tokens against MRP tokens. We note the best performance within each ablation and that overall the best performance in any of these experiments was achieved when including TVKT which is encouraging as they have to be included to be useful for pricing problems. CST appear to worsen performance which signals that there is little persistent leading information across channels for this problem. We exclude just using CST for the next ablation.

Table 3: Holdout RMSE results for module inclusion experiment.

| Measure | None | PatchTST | TVKT | CST | Both | DLinear |
|---------|------|----------|------|-----|------|---------|
| mean | **16.783** | 16.823 | 17.281 | 17.376 | 17.621 | 19.192 |
| std | **0.038** | 0.047 | 0.751 | 0.185 | 1.294 | 0.320 |
| min | 16.756 | 16.771 | **16.614** | 17.145 | 16.977 | 18.804 |

Table 4: Holdout RMSE results for scaling experiment.

| $d_m$ | Both | TVKT | None |
|-------|------|------|------|
| 32 | 17.462 | **16.555** | 16.847 |
| 64 | 17.431 | 16.803 | 16.720 |
| 96 | 16.848 | 17.034 | 16.680 |
| 128 | 16.884 | 17.056 | 16.646 |

### 5.2.3 Scaling

We ran a small scaling experiment with four levels of scaling $d_m \in \{32, 64, 96, 128\}$ and three module combinations: none, TVKT, and both. We avoided larger scales due to the small size of the dataset (approximately 17k series). The rule for other parameters was $d_{ff} = 2d_m$, the latent size in the mixer for CST is set as $d_{cross} = \frac{1}{4}d_m$, number of heads was selected as $\frac{1}{8}d_m$ and all models had two transformer blocks. The results are displayed in table 4. Interestingly TVKT performs better at a smaller latent size which may be explained by the size of the dataset. The small scale TVKT is also the best performing model of all in this experiment, tough this may not be consistently true due to previously observed variance in training models with auxiliary tokens. The other two ablations appear to scale with model size, though this effect for including both could be explained by gains in performance for MRP tokens as the ablation without any auxiliary information marginally improves with performance with scale.

### 5.3 Discussion

On the private dataset we demonstrate that our transformer architecture outperforms currently used in-house methods, DLinear, and PatchTST on a messy real-world problem. We ablate the model on a forecasting problem derived from a public dataset and demonstrate that our architecture is competitive with PatchTST and has better best-cases. We provide evidence that multiple resolution modelling outperforms single resolution patching. The experiments demonstrate that for certain forecasting problems modelling auxiliary data can add to generalisation contrary to common research trends, specifically the addition of TVKT which ended up producing the best performing model of all. More experimentation is needed to demonstrate the utility of the channel mixer and our suspicion is that it could be of value for the right channel structure.

## 6 Conclusion

The experiments provide clear evidence that our proposed MRT architecture works well on a noisy real-world problem. This is achieved with an architecture which includes both auxiliary variables which otherwise is often avoided in transformer architectures for time series forecasting. The modules designed for this purpose are novel as is the practice of passing all of their output tokens to a single attention mechanism. Our ablations show the fascinating effects different resolution sets have on performance and provide evidence in favour of multiple resolution approaches. One of the experiments shows a substantial improvement in performance over existing architectures. Given the difficulty of the underlying prediction problem this is a significant observation in favour of testing and potentially deploying transformer architectures on difficult real-world forecasting problems in the pricing domain and beyond.

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

# A   Additional Details About Experiments

## A.1   Real Markdown Problem

**Data Setting**   The main issue encountered when attempting to apply Multiple Resolution Tokenisation to pricing data is that the data has a distinct real-world flavour. Sales are aggregated within varying blocks of time that represent a duration of displayed price. The time series is irregularly spaced and each series has very few observations (at most 4). The data does not contain time series of sales when markdowns are not present. Each set of scan observations must be treated as its own series. Since only markdowns are observed, there is a clear selection effect as the process to select markdowns is biased toward existing more profitable strategies. This selection bias in the data generating process may result in poorer generalisation and overconfidence. The signal is also somewhat limited as there is only a narrow scope of markdown strategies that are regularly implemented. The dataset features many missing values and heuristics as a result of data acquisition problems.

**Auxiliary Information**   In the case of markdowns the inclusion of auxiliary data into the model is necessary as it is otherwise not possible to use it as a hypothesis generator for new pricing strategies. The decision variable (price) must be included as an input in the prediction model in order to be able to compare predictions under different markdown strategies. We try to predict both the full-price and the reduced price-sales concurrently, this defines the channels in $C$. This reflects the operational characteristics of the retailer where some proportion items that are about to expire get moved to a discounted section.

In the case of markdowns individual time series are relatively short and highly contextual. We take each series to be defined by as the unique combination of product, store, and expiration date. We want to aggregate all products of interest across all shops for all dates in a single model. We include auxiliary information because we do not believe that there is sufficient information in the starting course of a series to effectively differentiate amongst products and stores and expiration dates. We use following auxiliary information:

- Static:
    - Global: product group, brand, stock.
    - Specific: dummy variable for channel (either full price or reduced price sales channel).

- TVK:
    - Global: day of week, hour (half-hourly, continuous), month, which iteration of price reduction, proportion of stock marked down.
    - Specific: price (constant for full price prediction, changing for reduced price prediction).

### A.1.1   Preprocessing

From a model architecture standpoint the real markdown problem dataset poses two key problems: irregularly spaced sampling, and the varying length of the series.

**Quantisation**   The aggregation boundary times are rounded to the nearest 30 minute mark. The series is then quantised into 30 minute sections. We chose 30 minute intervals to create sequences with a non-trivial length, to add more auxiliary data to the sequence, and to enable a prediction of more potential pricing strategies. We quantise the entries in all columns which are aggregated (sales) over the given time period, the rest are simply copied. The quantisation is as simple as dividing the values by the number of 30 minute periods (exclusive of closure times) that occur from one price reduction to the next. There are two simplifications here: the rounding to 30 minutes means that we do not perform any imputation on the boundaries of the aggregations, the second is assigning the average value to each of the quantised fields. More disaggregated data would obviously work even better.

Note that there are approaches that use irregular time series natively such as neural differential equations (Kidger et al., 2020) to model the latent state between two observations. Unfortunately, an initial value problem has to be solved at every iteration which is computationally infeasible for large datasets.

**Coping with Varying Length**   To reduce the effect of changing sequence lengths we only keep the top five percent of sequences in terms of length. This also reduces the dataset to a much more manageable size. Since the sequences vary in the length of time, so does the number of steps obtained after quantisation. We fix the forecast horizon to the final 24 periods so $f = 24$. In its current design the model works for any fixed forecast horizon. We contend with varying input lengths by setting the lookback horizon to the longest sequence minus the fixed forecast horizon and padding all shorter sequences with values that result in zero vector tokens effectively deactivating that aspect of the network.

**Normalisation**   To further enable the aggregation of many potentially heterogeneous series and in line with effective practice for deep learning we use a collection of normalisation procedures. Normalisation is principally utilised to deal with the problem of different scales. Neural network training can easily result in unsatisfactory outcomes without scaling due to issues with the absolute and relative magnitude of gradients. Normalisation leads to improved generalisation due to increased stability in training and being able to deal with previously unseen scales. We employ two types of normalisation: instance normalisation for each series, and residual connections with normalisation in the transformer and channel mixer blocks. Instance normalisation subtracts the final value of the series from the entire series and optionally divides it by the standard deviation of the series and adds a learnable bias (affine transformation). The normalisation is reversed after the pass through the network. This type of normalisation has been shown to perform empirically and was originally proposed to combat distribution-shift/non-stationarity (Kim et al., 2021). The second type of normalisation is designed in a modular way and is applied after residual connections. It can either take the form of layer or batch normalisation. LayerNorm standardises the entire layer per instance, BatchNorm standardises each value in a layer across the batch. As opposed to language modelling, which typically uses LayerNorm, we use BatchNorm in experiments due to evidence that it is favourable in the time series context as it is better at handing outlier values (Zerveas et al., 2021). In preprocessing, we scale all continuous variables in the dataset. We fit many scalers based on what store type-product group combination a series falls into. The intuition is that this heuristic clustering should provide for more consistent scaling across different distributions of variables.

### A.1.2   Model details

Note that we use the same setting in the second experiment unless otherwise noted.

**Model Hyperparameters**   In this set of experiments we only used one set of model hyperparameters. The parameters picked reflect a set of informal observations on academic datasets and computational limits. We set the base latent dimension to $d_m = 64$, the dimension of the middle fully connected layer in the transformer $d_{ff} = 128$, the latent dimension in the channel mixer is set to $d_{cross} = 16$, we use the GeLU activation function, and set dropout to 0.0. We use 8 heads in the multi-head attention and stack 2 transformer layers. We compress the number of TVKTs and CST to 8. We pick the following set of resolutions $\{1, 2, 3, 4, 6, 8\}$. This results in $n_{MRP} = 24$, $n_{TVK} = 2 * 8$ (specific and global), $n_S = 4$, and $n_{CST} = 8$ for a total of 52 tokens. The resulting model had a parameter count of roughly $128k$.

**Training**   The batch size was set to 128, we used the Adam optimiser with a fixed learning rate of 0.0003, the loss function was minimising the root mean squared error (summed across channels, time steps, and samples). Note that MSE is not the optimal loss function for generalisation and calibration in this case, but investigating what is is beyond the scope of this work. We train the model for 20 epochs every experiment with early stopping if no improvement in validation score is observed for 3 epochs.

### A.2   Favorita

**Auxiliary Information**   We use following auxiliary information:

- Static:
  - Global: item class, item family, item id, is perishable
  - Specific: state, city, store id

- TVK:
  - Global: numerical distance to holidays, day of week, day of month, month
  - Specific: on promotion (analogous to pricing variable)

