# OpenReview forum: "Multiple-Resolution Tokenization for Time Series Forecasting with an Application to Pricing"
_TMLR — Rejected by TMLR_

### Review · Reviewer_Kv8x · 2024-07-16

**Summary Of Contributions:**

In short this paper's contributions can be summarised as follows:\
1.	The authors propose a multiple-resolution tokenisation approach for time series data that simultaneously processes information at different timescales.\
2.	They apply separate tokenisation modules for different types of auxiliary data (static variables, time-varying known variables), allowing explicit modelling of these inputs.\
3.	They employ a channel mixer module to extract cross-series information using a mixer architecture.\
4.	They propose a "reverse splitting" output head that scales favourably as the number of tokens increases.\
5.	They consider an application and evaluation on a real-world retail markdown pricing problem, showing competitive performance against internal models and outperforming some existing architectures.

**Audience:**

Yes

**Claims And Evidence:**

No

**Requested Changes:**

Some areas that the authors could consider adopting to improve the paper include:

1.	Conduct more extensive experiments across a wider range of data and timescales to better establish the model's robustness and generalizability, i.e. is the technique proposed tailored to this problem specifically or is it a general-purpose approach?
2.	Perform ablation studies to quantify the impact of each component (e.g. multiple-resolution tokenization, auxiliary data handling, channel mixer). This part lacks considerably.
3.	Potentially evaluate on public benchmark datasets in addition to the proprietary data to allow for better comparisons with existing methods; especially if the purpose of the paper is to also highlight the applicability of the proposed method to other areas and problems.
4.	Explore the model's scaling properties more systematically - how does performance change with model size, data size, etc.? This could be an extension of the experiments or could be added as part of a separate ablations section.

Some additional questions I've got:

1.	How sensitive is the model performance to the choice of resolutions in the multiple-resolution tokenisation?\
2.	What is the computational overhead of the proposed architecture compared to simpler approaches? Is the improved performance worth the added complexity?\
3.	Do the authors think that the approach could be extended to handle variable-length forecast horizons?

**Strengths And Weaknesses:**

Before going into details regarding the strengths and weaknesses I would like to highlight some moderate shortcomings that do not do the paper justice. For instance, the first paragraph of the introduction, in fact, the entire section, has no references at all. This is a bit problematic. In addition, the introduction seems to be jumping along multiple topics on stochasticity, non-stationarity, pricing models, etc. All these should have been elaborated and referenced upon.

Another glaring issue concerns the use of transformer models without a proper justification. It is true that we have seen several "foundation models for time series forecasting" lately, e.g. MOIRAI [1], but time series forecasting with transformers is not as obvious a solution as it is for Computer Vision. Some specific justification with respect to other approaches would have been beneficial.

Now in terms of the paper's strengths, these can be summarised as follows:\
The paper: \
1.	Addresses limitations of existing transformer approaches for time series by handling multiple resolutions and auxiliary data explicitly.\
2.	Provides a flexible architecture that can incorporate various types of inputs relevant to time series forecasting.\
3.	Demonstrates effectiveness on a challenging real-world problem with noisy data.\
4.	Proposes modules like the channel mixer and reverse splitting output head.\
5.	Shows Competitive performance against internal models.

Similarly for Weaknesses: \
The paper demonstrates\
1.	Limited experimental evaluation - only two main experiments on different weeks of data.\
2.	Lack of ablation studies to benchmark the impact of different components.\
3.	lack of reproducibility due to the use of proprietary dataset limits.\
4.	Computational constraints limited the scope of experiments and hyperparameter tuning.\
5.	That some hyperparameters were chosen based on informal observations rather than systematic tuning.

One more weakness that I would like to highlight separately is that as far as I can see the authors do not discuss the difference between the mixed-resolution tokenisation [2] and their multiple-resolution tokenisation approach. A couple of sentences on this would have been useful.

1. https://blog.salesforceairesearch.com/moirai/ \
2. https://arxiv.org/pdf/2304.00287

---

> ### Author Response · Authors · 2024-08-19
>
> Thank you for the comprehensive review. In response to requested changes:
> 1. Agreed that more experiments would be very beneficial in demonstrating that MRT is a general purpose model. The next iteration of this work will include tests on publicly available datasets to ensure reproducibility too.
> 2. Agreed, will be included in the next run of experiments. Was avoided due to resource constraints.
> 3. As stated in 1. we intend to evaluate the method on public benchmarks in the next iteration. We do note that given this model’s explicit focus on modelling auxiliary information it will be at best competitive on benchmarks like ETT for which it has been observed that the inclusion of auxiliary information induces overfitting. The proposed method should generalise to other problems where auxiliary information has a high degree of influence.
> 4. Agreed, very valuable ablation but beyond the resources available during this project, possibly an inclusion in a future iteration.
>
> In response to additional questions:
> 1. Not clearly established beyond more resolutions are better than a single resolution. Our hypothesis is that the scaling is monotonic and that more resolutions leads to improved outcomes on sufficiently large aggregated datasets. However, due to computational considerations picking a subset of resolutions is usually necessary.
> 2. The additional overhead is mainly reflected in the larger number of tokens passed to the self-attention which scales quadratically in the number of tokens. The number of tokens is very small compared to the context windows that modern LLMs are capable of (3-4 oom larger). The model is designed such that the total number of tokens passed to the transformer can be kept small. For time series with long lookback horizons the inclusion of high resolution patching will add a substantial number of tokens. This inclusion only makes sense when there is evidence of a substantial performance improvement which we would argue is problem dependent. More evidence is needed to justify additional complexity and to develop at least heuristics on how to set resolution sets.
> 3. Not trivially. One possibility would be to use the same padding solution applied to variable length inputs could be tried with variable length outputs using special tokens to reflect the forward padding. Our model architecture does not conform to autoregressive approaches as it can only iterate every least common multiple of the patch lengths.
>
> Other points:
> - We intend to add a justification for using transformers over other architectures. The appeal of the transformer was that it explicitly models pairwise relationships which we thought was important in highly auxiliary information driven processes such as pricing.
> - Our understanding of the distinction between mixed and multiple resolution approaches is that in mixed resolution approaches different patches in a single segmentation of an input can have different sizes. Whereas in multiple resolution approaches every segmentation is done using the same rule, but there are multiple segmentations each with its own rule. Given that our model includes all patch tokens in the same self attention mechanism we replicate aspects of a mixed resolution model. Every possible interaction of a mixed-resolution model is modelled explicitly in MRT.

---

> > ### Comment · Reviewer_Kv8x · 2024-08-19
> > **Revised Paper**
> >
> > Thanks for the responses provided to the reviews. I have read the other reviews as well as your responses and I think it would be helpful if you submitted a revised version of the paper to see some of the changes that have been requested. It would help if you highlighted them as well.

---

> > > ### Author Response · Authors · 2024-08-20
> > >
> > > Thank you for looking over our responses so quickly. Due to existing commitments and upcoming leave we cannot envision that the additional experiments will be completed before the second half of September. We can provide an updated version of the paper with highlighted changes by next week (with an in progress experiments section). Is this sufficient for now? We appreciate that it does not align with journal deadlines.

---

> > > > ### Comment · Reviewer_Kv8x · 2024-09-09
> > > > **New version**
> > > >
> > > > Thank you - I am anticipating the revised version of your paper.

---

> > > > > ### Author Response · Authors · 2024-09-26
> > > > >
> > > > > Apologies for the delay, we have uploaded a revised version.

---

> > > > > > ### Comment · Reviewer_Kv8x · 2024-10-10
> > > > > >
> > > > > > Thank you! I am about to submit my recommendation; can I check with you that tables 3 and 4 are new experiments?

---

> > > > > > > ### Author Response · Authors · 2024-10-10
> > > > > > >
> > > > > > > That is correct. In addition, figure 5 presents the results of experimenting with different resolution set combinations.

---

> > > > > > > > ### Comment · Reviewer_Kv8x · 2024-10-10
> > > > > > > >
> > > > > > > > Thank you - I am happy with that!

---

### Review · Reviewer_WhNa · 2024-07-30

**Summary Of Contributions:**

The work proposes a multi-resolution patching and tokenization approach towards time-series forecasting problem. Patches of time-series with different resolutions are turned into tokens. The proposed approach takes static and time-varying auxiliary information into consideration. It also adopts a cross-series mixture module to mix information across channels and proposes a reverse-splitting output head mirroring the multi-resolution patching of time-series data to output the prediction results. Experiment results on private data show competitive performance of the approach against PatchTST and DLinear.

**Audience:**

Yes

**Broader Impact Concerns:**

The work itself is a generic methodological approach. I do not have specific concerns about the work itself except for the reproducibility of experiment results but I would still encourage the author to briefly discuss broader and social impact concerns of the work.

**Claims And Evidence:**

Yes

**Requested Changes:**

I would like to see the following changes to the work:
1. Add discussions on the motivation of the output head design with reverse splitting and comparison with experiment results not regular output header without reverse splitting;
2. Include details on how DLinear and PatchTST models are adapted to accommodate auxiliary informations in the experiment setting of the work.
3. Clarify how a varying number of $n_{MRP}$ tokens, which depends on the length of time-varying know variable, can be projected into a set of fixed number of $n_{TKV}$ variables.

**Strengths And Weaknesses:**

Strength:
1. The work deals with a practical problem with real world significance.
2. Notations in the work are well defined with enough technical details.

Weakness:
1. Despite the practical value of the work, the original contributions of this work itself is not clear. Many essential technical component of the proposed approach including multi-resolution treatment to time series [1], tokenization of time-series patches [2], embedding of numerical values [3], and cross-series information mixing [4] are similar to existing works. The approach itself can be arguably viewed as a combination and extension of these works to a setting with auxiliary data and cross-series modelling.
2. To my best knowledge, the output head design with reverse splitting is novel. However, there's not enough motivation or study for this design itself.
3. The dataset in the experiment is neither a standard benchmark dataset nor a publicly available dataset. It is not clear how the experiment results of the work can be reproduced.
4. A minor notation issues. SKU in Page 2 right before Section 2.1 is not defined.

[1] Shabani, Amin, et al. "Scaleformer: Iterative multi-scale refining transformers for time series forecasting." arXiv preprint arXiv:2206.04038 (2022).

[2] Nie, Yuqi, et al. "A time series is worth 64 words: Long-term forecasting with transformers." arXiv preprint arXiv:2211.14730 (2022).

[3] Golkar, Siavash, et al. "xval: A continuous number encoding for large language models." arXiv preprint arXiv:2310.02989 (2023).

[4] Chen, Si-An, et al. "Tsmixer: An all-mlp architecture for time series forecasting." arXiv preprint arXiv:2303.06053 (2023).

---

> ### Author Response · Authors · 2024-08-19
>
> Thank you for your valuable review. In response to weaknesses:
> - To 1. We agree with the view that this can be seen as an extension as far as tokenisation is concerned. The patching of TVK variables using basis combinations is novel to the best of our knowledge, but it can also be seen as a derivative approach. From the tokenisation perspective our main contribution is passing all tokens to a single attention mechanism which is not present in other work where attention not used across resolutions.
> - To 3. Agree that this is a problem, we would like to test the model on a publicly available dataset in the next iteration of this work.
>
> In response to requested changes:
> 1. We will add motivation for the output head design. In short, the design is motivated by improved scaling, having a multiplicity of channels to learn processed tokens to output rules from, and symmetry with multiple resolution patching.
> 2. Neither DLinear nor PatchTST can accommodate auxiliary information by design. We should add a comparison with a model that can.
> 3. The number of tokens produced by the patching is independent of the length of the input sequence and only depends on values in the resolution set. The resolution set values define how many roughly equal parts for tokenisation a sequence should be split into the sum of which equals the number of tokens $n_{MRP}$. A forecasting problem $l=48$, $f=24$ for a resolution set value of 4 will after splitting produce 4 subsequences of length 12 for past data and 4 subsequences of length 18 for TVK.

---

### Review · Reviewer_tCcy · 2024-08-15

**Summary Of Contributions:**

This paper proposes a new transformer architecture tailored for time series forecasting, with a particular focus on time series tokenization. The main contributions of this work are the development of new tokenization modules that operate at multiple resolutions, a novel output head with favorable scaling properties, and the application of the proposed model to a real-world pricing prediction problem faced by a large retailer. The paper demonstrates the effectiveness of the proposed model through experiments on real-world markdown pricing data from a large retailer, where the model outperformed existing in-house methods and popular deep learning architectures like PatchTST and DLinear. The results suggest that the Multiple-Resolution Tokenization (MRT) architecture is particularly well-suited for complex, noisy, real-world forecasting problems, such as those found in the pricing domain.

**Audience:**

Yes

**Claims And Evidence:**

Yes

**Requested Changes:**

NA

**Strengths And Weaknesses:**

S:
- The paper presents an interesting application of transformer architecture for time series forecasting, particularly in the domain of pricing and sales prediction. The use of multiple-resolution tokenization is a significant advancement that addresses the challenges of capturing relevant information across different temporal scales simultaneously. This is a compelling approach that leverages the strengths of transformers in handling sequential data while addressing some of the inherent challenges in time series forecasting.

W:
- The paper does not provide sufficient introductory information about markdown pricing. For readers unfamiliar with this domain, it would be beneficial to include a brief overview of markdown pricing strategies, their importance in retail, and how they differ from other pricing strategies. Additionally, relevant citations should be added to guide newcomers to more detailed resources.
- The literature review is lacking in depth. The authors should expand this section to include a broader range of related work, particularly in the area of multi-resolution analysis and time series forecasting. Specifically, the discussion should include comparisons with neural operators and multi-fidelity models, as these methods address similar challenges in capturing information across different resolutions. References to works in neural operator and works in multi-resolution models, including should be included to provide context and highlight how the proposed method compares with existing approaches.
-- "Neural Operator: Learning Maps Between Function Spaces Nikola Kovachki, Zongyi Li, Burigede Liu, Kamyar Azizzadenesheli, Kaushik Bhattacharya, Andrew Stuart, Anima Anandkumar",
-- “Chen, Q., He, P., Yu, C., Zhang, X., He, J., and Li, Y. (2023). Multi-step short-term wind speed predictions employing multi-resolution feature fusion and frequency information mining. Renewable Energy, page 118942.”
-- “Doucoure, B., Agbossou, K., and Cardenas, A. (2016). Time series prediction using artificial wavelet neural network and multi-resolution analysis: Application to wind speed data. Renewable Energy, 92:202–211.”,
-- “Multi-resolution spatio-temporal prediction with application to wind power generation Zheng Dong, Hanyu Zhang, Shixiang Zhu, Yao Xie, Pascal Van Hentenryck”
- Given that neural operators are proposed to address similar problems of capturing multi-resolution information in time series, the authors should elaborate on the differences between their approach and neural operators. A discussion on why neural operators were not chosen for this particular application would add valuable insight. This comparison is critical as it would help in understanding the unique advantages and potential limitations of the proposed method.
- The idea of handling multi-resolution data is closely related to multi-fidelity models, which are used to capture correlations across multiple data resolutions. The authors should clarify how their approach differs from or complements these models. For instance, in multi-fidelity models, a surrogate model such as a Gaussian Process (GP) is often used to interpolate between different resolutions. The authors could discuss whether such an approach was considered, and if not, why their transformer-based method is preferable in this context.
- The experimental results are not robust enough to conclusively demonstrate the superiority of the proposed method. The results are limited in scope and do not explore a wide range of datasets or scenarios. This limitation makes it difficult to generalize the findings. The paper would benefit from additional experiments, including a more extensive comparison with other state-of-the-art methods, to strengthen the claims of effectiveness. Moreover, the discussion should include an analysis of why the proposed model performs better or worse in certain cases, particularly in relation to other methods like PatchTST and DLinear.

---

> ### Author Response · Authors · 2024-08-19
>
> Thank you for your thoughtful review. In response to the points under weaknesses:
> - We will add a short introduction on markdown pricing including references in the next iteration of the paper.
> - Thank you for bringing these works to our attention. Since the volume of approaches in this space is so vast we decided to narrow our focus on methods most similar to ours. Our work is chiefly different in three regards: explicit multi-resolution modelling of time-varying known variable representations, combining representations from all resolutions in a single attention mechanism, and using a novel output head which is more efficient. We will add some more references on multiple-resolution modelling.
> - Neural operators (NO) were not considered in the design stage of this model, so thank you for bringing them to our attention. The main advantage of NO in terms of capabilities is being able to process input output pairs at different resolutions in the same model. In our problem the resolution level of the sequence is the level of time aggregation, a fixed hyper-parameter related to the downstream decision task. We extract information at multiple resolutions from each sequence with the purpose of forecasting in one resolution, whereas NO builds a model in infinite dimensional space for all resolutions. The additional capability, while interesting, is not necessary as the predictive queries we are interested in have a fixed resolution. Given that the additional capabilities are not necessary, we prefer to use a relatively less complex architecture which avoids components such as integration.
> - The choice of the transformer as an interpolation engine was due to the explicit pairwise modelling of representations. We acknowledge that GPs are a powerful technique for multi-fidelity modelling, but the cubic computational complexity of GPs (scaling in number of samples) makes them unsuitable for large datasets such as the ones found in retail.
> - We agree that additional experiments are needed to conclusively establish that this method is superior. The limited scope is an unfortunate consequence of project limitations. We are planning to include experiments on public datasets in future iterations. However, large public datasets for long-horizon forecasting in which auxiliary information plays a significant role (like in pricing) are scarcely available. Most existing literature operates with the standard set of long-horizon datasets (ETT, and some combination of ECL, Exchange, Traffic, ILI, and Weather). It has been empirically found that adding auxiliary features or cross-series in some form on these datasets does not lead to improvements. This has resulted caused many downstream models to avoid modelling these important inputs as they are detrimental to results on these datasets. We believe that this is due to dataset bias as in a problem like pricing the inclusion of that information is obviously beneficial. So consequently we believe that on the standard set our model is at best competitive and instead demonstrate positive results for cases in an area where we suspected our model is superior. In pricing we think our model performs better as the complexity of the underlying process is high and highly dependent on auxiliary variables. Both PatchTST and DLinear do not model auxiliary variables, the former extracts representations in one resolution and the later learns a simple model on compressed measures of the sequence at multiple resolutions. We appreciate that a comparison should have been included with a model which models auxiliary variables and hope to establish the degree of impact in a set of ablations which we will perform for the next iteration.

---

### Decision · Action_Editor_tiLz · 2024-11-04

**Recommendation:** Reject

**Comment:**

The submission addresses the impact of tokenization with transformers for time series forecasting in the pricing domain. The authors propose a multi-resolution tokenization module that handles multiple resolutions and auxiliary data explicitly. The technical contributions include a multi-resolution module, a mixer-based component for merging known and auxiliary variables, and an output head designed to improve scalability. Experiments were initially conducted on proprietary pricing prediction datasets provided by a large retailer.

The paper received mixed initial feedback. Reviewers appreciated the idea of adapting tokenization for time series forecasting with transformers. However, they also raised concerns about the paper’s presentation, the lack of contextual positioning regarding related work (e.g., neural operators, multi-fidelity and multi-scale transformers), and the experiments, which were limited to a single proprietary dataset and lacked comparative studies and ablations to validate the proposed approach. During the discussion period, the authors revised the paper structure and added ablations on a public dataset. Although reviewer Kv8x was satisfied with these changes, reviewers tCcy and WhNa felt that concerns around positioning and experimental rigor were still insufficiently addressed, leading them to recommend rejection.

The AE has thoroughly reviewed the submission and discussion. While the AE acknowledges that the problem of tokenization in multi-resolution transformers for time series forecasting is significant, they also believe that substantial improvements are required. The paper’s presentation should be refined; for example, tables and figures should be self-contained. Additionally, the paper’s technical contributions need clearer articulation, stronger positioning within the literature, and more systematic experimental validation. Therefore, the AE recommends rejection.

**Audience:**

The paper addresses time series forecasting and the impact of multi-resolution tokenization with transformers, which is likely to attract the interest of a broad TMLR audience.

**Claims And Evidence:**

The claims are not fully supported by evidence: the primary experiments are restricted to a single proprietary dataset, the added ablations studies on a public dataset are incomplete, and the paper lacks comparison and positioning with respect to related works.

**Resubmission Of Major Revision:**

The authors may consider submitting a major revision at a later time.